# Salvage Radiotherapy Plus Androgen Deprivation Therapy for High-Risk Prostate Cancer with Biochemical Failure after High-Intensity Focused Ultrasound as Primary Treatment

**DOI:** 10.3390/jcm11154450

**Published:** 2022-07-30

**Authors:** Ying-Che Huang, Chih-Hsiung Kang, Wei-Chia Lee, Yuan-Tso Cheng, Yao-Chi Chuang, Hung-Jen Wang, Fu-Min Fang, Po-Hui Chiang

**Affiliations:** 1Department of Urology, Kaohsiung Chang Gung Memorial Hospital and Chang Gung University College of Medicine, Kaohsiung 83301, Taiwan; fireman21316@hotmail.com (Y.-C.H.); chkang5801@gmail.com (C.-H.K.); wegar@cgmh.org.tw (W.-C.L.); ytcheng@cgmh.org.tw (Y.-T.C.); chuang8270@cgmh.org.tw (Y.-C.C.); wang6107@cgmh.org.tw (H.-J.W.); 2Department of Radiation Oncology, Kaohsiung Chang Gung Memorial Hospital and Chang Gung University College of Medicine, Kaohsiung 83301, Taiwan; 3Department of Medicine, Chang Gung University College of Medicine, Taoyuan 33302, Taiwan; 4College of Medicine, Kaohsiung Medical University, Kaohsiung 80708, Taiwan; 5Jhong Siao Urological Hospital, Kaohsiung 800, Taiwan

**Keywords:** androgen antagonist, high-intensity focused ultrasound ablation, prostatic neoplasms, radiotherapy, treatment failure

## Abstract

We conduct a retrospective analysis of salvage radiotherapy plus androgen deprivation therapy (SRT+ADT) for high-risk prostate cancer patients with biochemical failure after high-intensity focused ultrasound (HIFU) as the primary treatment. A total of 38 patients, who met the criteria of biochemical failure and were consecutively treated with SRT+ADT, were enrolled. All patients received intensity modulated radiotherapy with a median dose of 70 Gy to the clinical target volume. ADT was given before, during or after the course of SRT with the duration of ≦6 months (n = 14), 6–12 months (n = 12) or >12 months (n = 12). The median follow-up was 45.9 months. A total of 10 (26.3%) patients had biochemical failure after SRT+ADT. The cumulative 5-year biochemical progression free survival (b-PFS) and overall survival (OS) rate was 73.0% and 80.3%, respectively. A nadir prostate-specific antigen (nPSA) value 0.02 ng/mL was observed to predict the b-PFS in multivariate analysis. The 5-year b-PFS was 81.6% for those with nPSA < 0.02 compared with 25.0% with nPSA ≧ 0.02. The adverse effects related to SRT+ADT were mild in most cases and only three (8%) patients experienced grade 3 urinary toxicities. For high-risk prostate cancer after HIFU as primary treatment with biochemical failure, our study confirms the feasibility of SRT+ADT with high b-PFS, OS and low toxicity.

## 1. Introduction

According to the nationwide health and welfare database, the incidence of prostate cancer in Taiwan has been increasing and became an important health challenge in the past couple of years [1]. Based on the Gleason Score, PSA level and clinical stage, patients with prostate cancer can be categorized into different risk groups. Traditionally, radical prostatectomy or radiotherapy (RT) combined with hormone therapy is the established and definitive treatment for high-risk prostate cancer. However, some patients are not suitable for the major operation or cannot tolerate the radiation-related complications, which may cause a great impact on the patients’ quality of life. In recent years, considering the acceptable oncological control and limited morbidity, high-intensity focused ultrasound (HIFU) has been used as a noninvasive primary therapy for patients with localized prostate cancer. HIFU uses a focused ultrasound wave that mechanically and thermally induces tissue damage, which causes coagulative necrosis through tissue cavitation and temperature elevation [2]. The whole gland ablation by HIFU for patients with localized (stage T1c to T3) prostate cancer, classified as the high-risk group based on D’Amico classification, has been proved to be a potentially effective and safe modality with acceptable disease-specific mortality and treatment-related morbidity in the literature [3] and the experience of our institute [4,5,6,7]. The 5-year biochemical failure rate treated by HIFU for high-risk prostate cancer was reported around 32–50% [3,4]. However, the ideal treatment option for those patients with local relapse after HIFU as primary therapy remains unclear. Previous studies have demonstrated that salvage RT (SRT) provides encouraging results after HIFU failure, and some of the patients may receive additional androgen deprivation therapy (ADT) [8,9,10]. To our knowledge, neither subset analysis on the high-risk group nor the well-documented outcome of SRT plus androgen deprivation therapy (SRT+ADT) has been performed so far. We herein examined and analyzed the outcome of SRT+ADT in high-risk prostate cancer patients after HIFU failure. We aim to identify a satisfactory oncologic treatment that can be used as a guide for men with high-risk prostate cancer after HIFU failure to choose safely.

## 2. Materials and Methods

### 2.1. Patient Cohort

This is a retrospectively observational study at Kaohsiung Chang Gung Memorial Hospital. From December 2009 to July 2019, 405 patients with prostate cancer underwent whole-gland HIFU at our institution. The criteria for HIFU in the institute were (1) localized prostate cancer with stage cT1-3N0M0, and (2) patients are unsuitable candidates for surgery or reluctant to receive RT. The HIFU device was Ablatherm^®^ Integrated Imaging (EDAP TMS SA, Vaulx-en-Velin, France). Of these 405 patients, 76 underwent SRT and of these, 49 received additional ADT with a minimum of 1-year follow-up. Among these 49 consecutive patients, 38 who were categorized as the high-risk group before HIFU were selected for analysis.

Before HIFU, all patients underwent digital rectal exam (DRE), PSA test, prostate biopsy, pelvic multi-parametric magnetic resonance imaging (MRI) or computed tomography (CT) and bone scan for preoperative staging. Patients were stratified into low-, intermediate- and high-risk groups according to the D’Amico classification. After HIFU, the PSA level is monitored every 3 months based on our institutional protocol. The definition of biochemical failure after HIFU treatment was either the nadir prostate-specific antigen (nPSA) value plus 2 ng/mL (Phoenix definition) or PSA increasing on two consecutive measurements. Prostate biopsy was optionally arranged but image studies for clinical re-staging, including MRI, CT, and bone scan, were routinely preformed due to biochemical failure. If biochemical recurrence is detected and no evidence of distant metastasis, in this study population of high-risk prostate cancer, SRT+ ADT is the standard strategy. This study was conducted by the Declaration of Helsinki and relevant guidelines and approved by the institutional review board of Chang Gung Memorial Hospital, Taiwan (No: 202001472B0).

### 2.2. Salvage Radiotherapy

All patients received SRT by using the technique of intensity modulated technique (IMRT). The technique of IMRT for prostate cancer in the institute has been published [11]. Sequential two treatment phases were given. In phase I, the clinical target volume (CTV) comprised the prostate and seminal vesicles without (n = 13) or with (n = 25) the pelvic lymph node. The decision of elective pelvic lymph node irradiation was basically according to the Roach’s formula [12]. The prescribed dose was 1.8–2.0 Gy per fraction per day to achieve 44–50 Gy at phase I. In phase II, the CTV was reduced to focus on the prostate and gross tumor area to escalate the received total dose to a median dose of 70 Gy (range 60–72 Gy). Seven (18.4%) patients received less than 70 Gy; five of them received 66–68 Gy because no tumor could be found from MRI, and one patient refused further irradiation after 60 Gy for personal reason. The CTV was expanded three-dimensionally with a 5–10 mm margin to build up the planning target volume (PTV). A total of 30 (79%) patients also underwent a supplemental image-guided technique to reduce setup uncertainty. The normal organs (e.g., rectum, urinary bladder and small bowels) located at the PTV level on the CT images were contoured for dose constraints. Each IMRT plan adopted 6–8 coplanar portals delivered by a 6–10 MV linear accelerator. The planning system was Pinnacle3 (Philips Medical Systems, Madison, WI, USA). The prescribed dose must cover 95% of the PTV, and the small hot spot area over 110% of the prescribed dose should be limited to within the PTV. The surrounding organ dose–volume constraints were usually as follows: the relative volume of rectum receiving dose over 70 Gy should not be more than 25%; the max dose in the small bowel should be limited to 55 Gy. As for the urinary bladder, at least one-half of the volume kept the dose no more than 65 Gy.

### 2.3. Androgen Deprivation Therapy

All patients received ADT before, during or after the course of SRT. LHRH-agonists (Leuprorelin or Goserelin) or LHRH-antagonists (Degarelix) were given. The time to terminate ADT was usually when two consecutive undetectable PSA could be reached or empirically decided by the individual urologist. We categorized the duration of ADT prescribed into three groups: ≦6 months (n = 14), 6–12 months (n = 12), and >12 months (n = 12). 

### 2.4. Endpoints

Our institutional protocol for follow-up after SRT+ADT is based on the 3-monthly postoperative PSA level. The primary endpoint was biochemical-progression free survival (b-PFS), defined as no biochemical relapse (PSA value under nadir plus 2 ng/mL) and without re-administration of ADT and overall survival (OS). The secondary endpoint was adverse events (AEs). The Common Terminology Criteria for Adverse Event (CTCAE) v4.0 was used to evaluate the tolerance of SRT+ADT after HIFU. AEs were categorized into 5 grades as follows: grade 1, mild; grade 2, moderate; grade 3, severe or medically significant; grade 4: life-threatening or urgent intervention indicated; and grade 5, death related to AE. Acute AEs are defined as those occurring within 3 months; late AEs are those occurring later than 3 months.

### 2.5. Statistical Analysis

The duration of survival was calculated from the first day of the treatment (SRT or ADT) to the last day of follow-up or patient death. Kaplan–Meier method was used to build survival curves and a log-rank test was performed to analyze the survival difference between groups. The variables, including age, initial T status, GS, PSA prior HIFU, recurrent T status, PSA prior SRT, duration of ADT, nPSA value post SRT+ADT, and time, to achieve nPSA were separately analyzed for their prognostic value of b-PFS and OS. The correlation between nPSA and treatment failure was analyzed and the cut-off of nPSA was further identified using ROC analysis and Youden function [13,14]. Those variables with *p*-value < 0.2 in univariate survival analysis were chosen into the multivariate analysis using the Cox proportional hazards model. All p values reported are two-sided, with a *p*-value of <0.05 regarded statistically significant. The softwares, Microsoft SPSS version 22 and Stata version 15.1 (Stata Corporation LP, College Station, TX, USA), were used for data processing.

## 3. Results

### 3.1. Patient Characteristics

Relevant patients’ characteristics are shown in Table 1. Among the 38 patients, 32 received 1 HIFU session, 5 received 2 sessions and 1 received 3 sessions. The median PSA value at initial diagnosis and biochemical failure were 23.73 (range 1.44–120) and 2.76 (range 0.4–57.83), respectively. For the 32 patients receiving multiparametric MRI after biochemical failure, all of them presented with abnormal image findings meeting the criteria of tumor recurrence. The recurrence T and N status based on the AJCC 8th edition were rcT2: 23 (60%) cases, rcT3: 8 (21%) cases, rcT4: 1 (3%) case, rcN0: 30 (79%) cases, rcN1:2 (5%) cases, and unknown: 6 (16%) cases, respectively. No distant metastasis was observed before the salvage treatment. The median interval between the last HIFU treatment and ADR+SRT was 11.2 months (range 1–42.6). The median follow-up was 45.9 months (range 20.1–93.1).

### 3.2. Survival Outcome

10 (26.3%) patients had biochemical failure after SRT+ADT. The median and mean time to disease progression were 21.2 months (range 6.3–36.4) and 20.6 months (SD 9.52), respectively. After further hormone therapy and/or chemotherapy, six of them were alive with disease free, two alive with the disease, and two died of lung or bone metastasis. The cumulative 5-year b-PFS and OS of the cohort were 73.0% and 80.3%, respectively (Figure 1). A statistically significant trend (r = 0.527, *p* = 0.001) was observed that those patients with a higher nPSA value after SRT+ADT had a higher probability of failure of the salvage treatment. The median time to achieve nPSA was 8.1 months (range 2.2–22.5 months). The 5-year b-PFS and OS rates among the three groups of patients receiving various ADT duration revealed no statistical difference (Figure 2). An nPSA cut-off value of 0.02 ng/mL was observed to predict the b-PFS and OS. The 5-year b-PFS and OS rates were statistically significantly higher for those with nPSA < 0.02 (b-PFS: 81.6% vs. 25.0%, *p* < 0.001; OS: 90.4% vs. 37.5%, *p* = 0.04), compared with those with nPSA ≧ 0.02 (Figure 3). After the multivariate Cox model analysis, we observed that nPSA < 0.02 was the only significantly favorable predictor of b-PFS (Table 2).

### 3.3. Adverse Effects

Urinary AEs include urinary retention, incontinence, urgency, obstruction, and hematuria. Gastrointestinal (GI) AEs include hemorrhage, ulcer, obstruction of the bowel, and diarrhea or constipation. Concerning the AEs of SRT, minor urinary and GI disorders were common (Table 3). For acute AEs, 20 (53%) and 3 (8%) patients experienced grade 1 and 2 urinary toxicities; 23 (61%) and 1 (3%) patients experienced grade 1 and 2 GI toxicity, respectively. No grade 3 or above acute AE was reported. For late AEs, five (13%) patients experienced grade 2 urinary toxicities, and three (8%) patients experienced grade 3 urinary toxicities, which needed optical internal urethrotomy to relieve severe urethral stricture. No minor or severe late GI toxicities were reported. The AEs of ADT were also rare and mild. In total, two (6%) patients and one (3%) patient experienced grade 1 and grade 2 toxicity, respectively. The AEs included hot flash (one patient), hyperhidrosis (one patient), and skin hyperpigmentation (one patient). 

## 4. Discussion

Although still under evaluation and controversial, the feasibility and the oncological outcomes with HIFU have been demonstrated in the treatment of localized prostate cancer in several studies [3,15]. The main limitation of HIFU is the lack of long-term follow up and limited number of cases, which makes this noninvasive primary therapy remain controversial. However, some have reported encouraging the outcome with longer follow-up, including high cancer-specific survival and metastasis-free survival in 6.4 years of median follow up [3]. 

It has been reported that 27% of patients was identified as local recurrence and will be, therefore, managed by active surveillance, palliative hormonal treatment, repeat HIFU, salvage RP or SRT [3]. Active surveillance and hormonal treatment alone are not curative. Repeat HIFU may increase the risk of complications and the oncological outcome for high-risk patient is unclear. Although the potential of salvage RP after HIFU has been reported in small groups of patients [16], many HIFU patients are not candidates for radical surgery. Earlier studies have demonstrated the feasibility of SRT+ADT after HIFU failure [8,9,10]. However, there remains a paucity of studies on the potential of SRT+ADT in high-risk prostate cancer patients after HIFU failure. On the basis of oncological outcome and acceptable AE, our study suggested SRT+ADT can be considered a treatment option after HIFU failure.

Currently, there is no firm recommendation regarding clinically relevant PSA relapse threshold post-HIFU, although the increasing use of biopsy-proven local recurrence was reported in previous studies [8,9,17]. The usual Stuttgart definition of biochemical failure for the patients treated with HIFU may underestimate the result because the prior study only included those patients with low- and intermediate-risk prostate cancer [18]. Given our study focusing on high-risk prostate cancer post HIFU, PSA level increasing on two consecutive measurements with imaging of local recurrence is also defined as failure. Moreover, little is known about the best treatment option for those patients with prostate cancer relapses after HIFU. In our study, SRT+ADT after HIFU as primary treatment provides a satisfactory oncological outcome. The 5-year b-PFS and OS were 73.0% and 80.3%. We did not find a significant survival difference in the duration of ADT for b-PFS or OS, which might be related to the inhomogeneous ADT regimen. From the previous randomized trials on the SRT+ADT after RP for prostate cancer, the addition of ADT may improve outcomes for men receiving SRT, especially for those patients with higher PSA level [19,20]. However, the optimal type of ADT and for how long in addition to the SRT remained unclear.

As for patients accepting HIFU as primary treatment, only a few studies provided evidence of the efficacy of SRT alone after HIFU. In Rivière et al.’s study, 83 (out of 100) patients received SRT alone post-HIFU for recurrent localized prostate cancer [9]. The 5-year b-PFS was 72.5% (b-PFS defined as no three consecutive rises in PSA with a velocity of 0.4 ng/mL per year or PSA > 11.5 ng/mL and no additional treatment). Among these 83 patients, 14 were high-risk and 5-year b-PFS was only 55%. The other 17 (out of 100) patients receiving SRT+ADT were excluded for further analysis. In Pasticier et al.’s study, 32 (out of 45) patients received SRT post HIFU for recurrent localized prostate cancer [8]. The 5-year b-PFS was 64% (b-PFS defined as PSA < 1.5 μg/L and not increasing on two consecutive measurements, or no ADT). Among these 32 patients, 11 were high-risk but no risk stratification survival data provided. The other 13 (out of 45) patients receiving SRT+ADT were also excluded for further evaluation. In Munoz et al.’s study, 24 patients received SRT pos-HIFU for recurrent localized prostate cancer [10]. The 3-year b-PFS was 77.78% (b-PFS defined as PSA < nPSA + 2 ng/mL). Among the 24 patients, 2 were high risk and 17 received additional ADT. Given the small sample size, they did not provide a sub-analysis on different risks or different therapy groups. To our knowledge, no studies focus on SRT+ADT for recurrent prostate cancer after HIFU as primary treatment, especially for high-risk patients. For the 38 high-risk patients treated with SRT+ADT in our study, b-PFS is 73.0% at 5 years, which is higher than prior studies treated with SRT alone. 

Previous studies suggest a correlation between nPSA post SRT and biochemical failure. However, an ideal nPSA cut-off point to separate “likely” and “less-likely” biochemical failure varies from different studies. In our study, we observed nPSA less than 0.02 ng/mL post-SRT+ADT was a significantly favorable indicator of b-PFS. Rivière et al. concluded that nPSA was a strong prognostic factor of b-PFS, with a 5-year b-PFS of 87% if an nPSA of ≤0.2 ng/mL was reached after SRT [9]. Munoz et al. reported that patients who achieved an nPSA ≤ 0.35 ng/mL were more likely to experience freedom from biochemical failure than those who did not after SRT (5-year b-PFS: 87.7 vs. 50%, respectively, *p* = 0.01) [10]. Other previous studies on patients with prostate cancer treated with first-line RT also found a similar correlation between biochemical failure and nPSA and a wide nPSA cut-off point range from 0.2 to 2.0 ng/mL [21,22,23]. However, in these studies, the choice of an appropriate nPSA cut-off point only considers the effect of RT and does not take the effect of ADT into account. 

Regarding the AEs of SRT after HIFU, different studies use different grading systems. We used the CTCAE v4.0 grading system and most of the AEs are mild (grade 1). Rivière et al. used the CTCAE v3.0 grading system, which reported about 10% of patients presented with grade 3 urinary AEs [9]. Munoz et al. used the Radiation Therapy Oncology Group (RTOG) grading system to evaluate bladder and bowel toxicity. Only 4.2 % of patients presented with late grade ≥ 3 GI toxicity, and 12–16% of patients presented with acute and late grade ≥ 3 GU toxicity [10]. Compared to those previous studies, our result shows slightly lower toxicity of SRT after HIFU no matter of GI or GU aspect. It might be related to the advanced technique of radiotherapy applied in our patients. However, stricture recurrence is common for SRT following HIFU. Three cases with urethral strictures in our study were successfully managed by optical internal urethrotomy without urethroplasty. Long-term follow-up is still required for late radiation toxicity.

Regarding the AEs of ADT, the overall hormone toxicity rate was 7% in our series. Compared with the RT alone group, patients in the endocrine plus RT group were reported to have more urinary incontinence, urgency and urethral stricture [24]. Some studies also indicated that ADT increases the incidences of cardiovascular disease [25] and bone fractures [26]. However, in Asian people, the incidence of ischemic heart disease and bone fracture after ADT is much lower compared with Caucasian people [27]. We did not find an elevated incidence of cardiovascular disease or bone fracture, which might be related to race and the short-term use of ADT. 

Several limitations exist in our study. First, even though it is the largest study on patients with high-risk prostate cancer after HIFU, the sample size remains relatively small. Second, even though all the patients in this study underwent SRT+ADT, the ADT regimen is inhomogeneous, and our outcomes and adverse effects can be over/underestimated. Lastly, our study cohort only includes patients with high-risk prostate cancer. Hence, the findings may not be extrapolated to patients with less aggressive prostate cancer.

## 5. Conclusions

For high-risk prostate cancer after HIFU as primary treatment with biochemical failure, our study suggests the feasibility of SRT+ADT with high b-PFS, OS and relatively low AEs. Further prospective studies with larger sample sizes are needed to confirm our result.

## Figures and Tables

**Figure 1 jcm-11-04450-f001:**
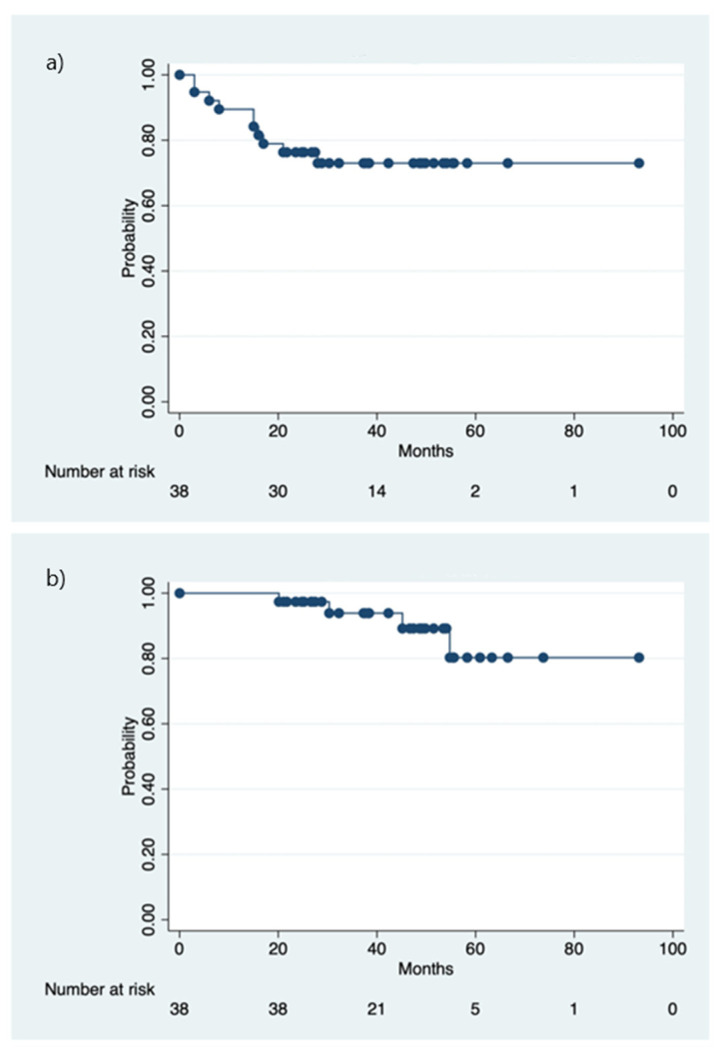
The cumulative 5-year biochemical-progression free survival (**a**) and overall survival (**b**).

**Figure 2 jcm-11-04450-f002:**
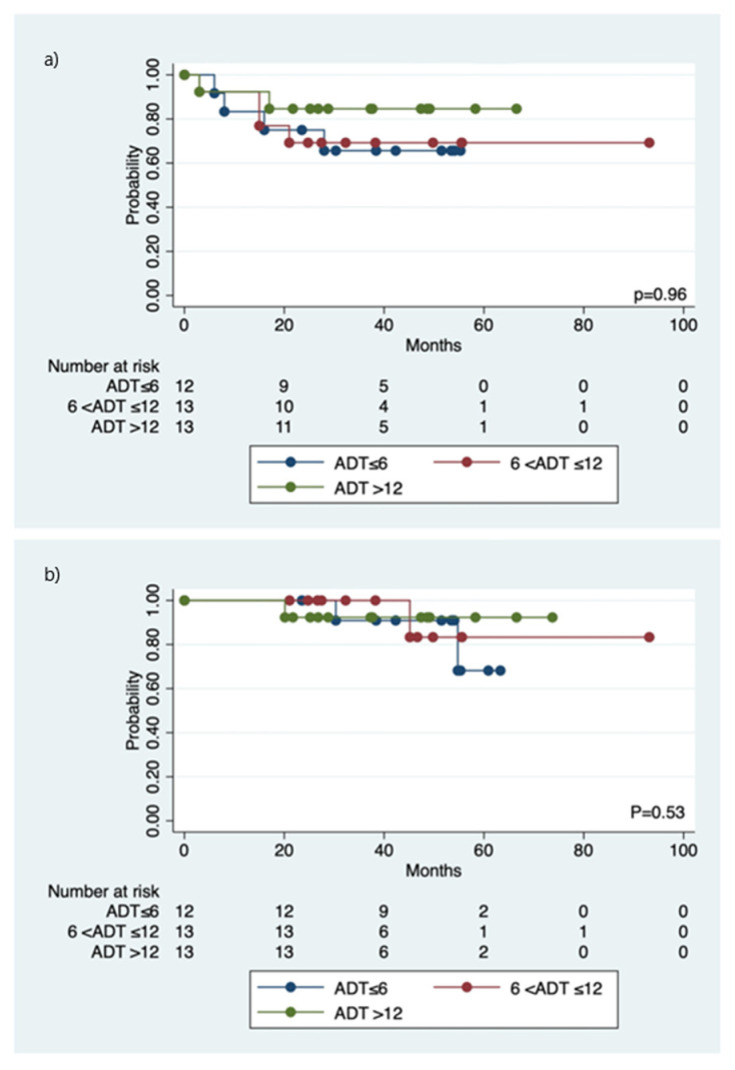
The cumulative 5-year biochemical-progression free survival (**a**) and overall survival (**b**) between the various ADT duration.

**Figure 3 jcm-11-04450-f003:**
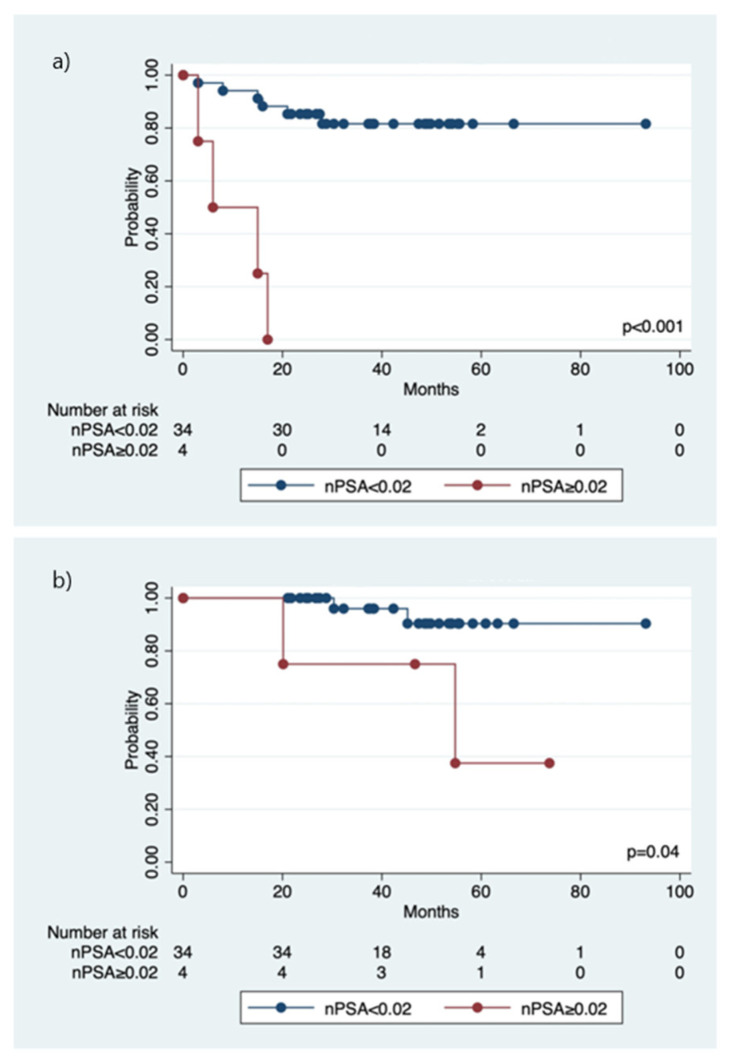
The cumulative 5-year biochemical progression free survival (**a**) and overall survival (**b**) for those with nadir PSA (nPSA) < 0.02 versus nPSA ≧ 0.02.

**Table 1 jcm-11-04450-t001:** Patient characteristics (n = 38).

Parameter	Number
Age, median (range), year	68.5 (55–86)
PSA, median (range), ng/mL	
Prior HIFU	23.73 (1.44–120)
Prior SRT	2.76 (0.4–57.83)
Number of HIFU, %	
1	32 (84)
2	5 (13)
3	1 (3)
Initial T status, %	
cT1	8 (21)
cT2	22 (58)
cT3	8 (21)
Initial N status, %	
N0	38 (100)
Gleason score, %	
≤6	3 (8)
7	13 (34)
≥8	22 (58)
Months between HIFU and SRT, median (range)	11.2 (1–42.6)
Duration of ADT, months	
ADT ≤ 6	14 (37)
6 < ADT ≤ 12	12 (31.5)
ADT > 12	12 (31.5)

**Table 2 jcm-11-04450-t002:** Cox regression model to predict variables associated with biochemical progression free survival and overall survival.

	Biochemical-Progression Free Survival				Overall Survival			
	UVA	MVA			UVA	MVA		
	*p*	HR	95% CI	*p*	*p*	HR	95% CI	*p*
Age: ≦68 vs. >68 years	0.14	1.21	0.23–6.50	0.82	0.29			
Initial T status: T1-2 vs. T3	0.95				0.90			
Gleason score: ≦7 vs. >7	0.13	1.80	0.32–10.02	0.50	0.39			
PSA prior HIFU: ≦23.73 vs. >23.73	0.54				0.97			
Recurrent T status: T1-2 vs. T3-4	0.41				0.79			
PSA prior SRT: ≦2.76 vs. >2.76	0.14	1.35	0.26–7.10	0.72	0.14	2.87	0.16–52.21	0.47
Duration of ADT: ≤6 m v. 6–12 vs. >12 m	0.35				0.66			
nPSA post SRT+ ADT: <0.02 vs. ≥0.02	<0.001	9.35	1.54–56.86	0.01	0.04	3.18	0.24–42.14	0.38
Time to achieve nPSA: ≤8.6 m vs. >8.6 m	0.97				0.36			

UVA = Univariable analysis, MVA = Multivariate analysis.

**Table 3 jcm-11-04450-t003:** Adverse events related to SRT or ADT.

Adverse Events	Grade 1 n (%)	Grade 2 n (%)	Grade 3 n (%)
SRT related, acute			
Urinary	20 (53)	3 (8)	-
Gastrointestinal	23 (61)	1 (3)	-
SRT related, late			
Urinary	1 (3)	5 (13)	3 (8)
Gastrointestinal	-	-	-
ADT related			
Hot flush	1 (3)	-	-
Hyperhidrosis	1 (3)	-	-
Skin hyperpigmentation	-	1 (3)	-

SRT: salvage radiotherapy; ADT: androgen deprivation therapy; Grading was based on CTCAE v.4; acute adverse events were those occurring within 3 months and late were those occurring 3 months later. Urinary adverse events include urinary retention, incontinence, urgency, obstruction, hematuria and erectile dysfunction. Gastrointestinal adverse events include hemorrhage, ulcer, obstruction of the bowel, and diarrhea or constipation.

## Data Availability

The data that support the findings of this study are available on request from the corresponding author. The data are not publicly available due to privacy or ethical restrictions.

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
