# Peer review of "Salvage Radiotherapy Plus Androgen Deprivation Therapy for High-Risk Prostate Cancer with Biochemical Failure after High-Intensity Focused Ultrasound as Primary Treatment"

_jcm, 2022, doi:10.3390/jcm11154450_

Round 1

Reviewer 1 Report

The authors evaluated the efficacy of salvage radiotherapy (SRT) plus ADT for high-risk prostate cancer with biochemical failure after HIFU treatment. The results are interesting and informative. However, the manuscript needs several major revisions.

1.       Page 3, line 110. The patients included various ADT duration of <6months, 6-12 months and >12 months. The authors evaluated the biochemical PFS from the last day of SRT. The biochemical PFS depends on the timing to stop ADT. The results should be evaluated separately in each group of ADT duration.

2.       Page 4, line 160. The authors evaluated the OS and biochemical PFS in nadir PSA of <0.02 and >0.02. The nadir PSA under ADT is a parameter to see the efficacy of ADT not the efficacy of SRT.

3.       Page 7, line 175. The authors showed the rate of adverse effects by SRT after HIFU. They might be the rate of acute adverse effects. The late adverse effects is more important in this study. The authors should make discussion.

4.       Figure 1, 2. The number at risk should be described in the figures.

Author Response

The authors evaluated the efficacy of salvage radiotherapy (SRT) plus ADT for high-risk prostate cancer with biochemical failure after HIFU treatment. The results are interesting and informative. However, the manuscript needs several major revisions.

  1. Page 3, line 110. The patients included various ADT duration of <6months, 6-12 months and >12 months. The authors evaluated the biochemical PFS from the last day of SRT. The biochemical PFS depends on the timing to stop ADT. The results should be evaluated separately in each group of ADT duration.

Response: Thanks for the insightful comment. We have added the results of 5-year b-PFS and OS rates among the three groups receiving different ADT duration in the revised manuscript. As shown in Figure 2, no statistically significant difference was observed.

  1. Page 4, line 160. The authors evaluated the OS and biochemical PFS in nadir PSA of <0.02 and >0.02. The nadir PSA under ADT is a parameter to see the efficacy of ADT not the efficacy of SRT.

Response: Thanks for the insightful question. We agree that the nadir PSA under ADT is a parameter to see the efficacy of ADT not the efficacy of SRT. All our patients were treated with ADT+SRT, thus, we could not tell whether the synergistic effect of the ADT+SRT exists on the survival outcome. Nevertheless, as shown in paragraph 4 of Discussion, we have stressed that in our study, nPSA less than 0.02 ng/ml post SRT+ADT was a significantly favorable indicator of b-PFS, which is lower than those in previous studies.

  1. Page 7, line 175. The authors showed the rate of adverse effects by SRT after HIFU. They might be the rate of acute adverse effects. The late adverse effect is more important in this study. The authors should make discussion.

Response: Thanks for the kindly comment. As shown in Table 3, the acute and late adverse effects related SRT or ADT were clearly revealed and stated. However, long-term follow-up is still required for late radiation toxicity. We added some discussion regarding this issue in paragraph 6 of Discussion, which is “Compared to those previous studies, our result shows slightly lower toxicity of SRT after HIFU no matter of GI or GU aspect. It might be related to the advanced technique of radiotherapy applied in our patients. However, stricture recurrence is common for SRT following HIFU. Three cases with urethral strictures in our study were successfully managed by optical internal urethrotomy without urethroplasty. Long-term follow-up is still required for late radiation toxicity.”

  1. Figure 1, 2. The number at risk should be described in the figures.

Response: Thanks for the kindly comment. We have added the number at risk in the figures.

Reviewer 2 Report

Congratulations to the authors on your present work. It is very clear that you have a wide experience with whole gland HIFU and that you have explored the limits of this technique, which is admirable. I think the present manuscript has a lot of potential but needs to be refined, moderated, and be very careful with the description of the outcomes, variables and results in order to avoid confusions to the readers. Here are some recommendations for you to consider:

·      Please reference Roach´s formula for pelvic node irradiation

·      You describe your PSA follow up with measurements every 3 months, how long is that frequency sustained for?

·      When you mention a median follow up of 45.9, is that after the radiation? Or are you counting since the last HIFU?

·      How was it decided to go for sRT rather than surgery for these patients?

·      I would describe what was the reason to choose HIFU for a high-risk patient. Most of the recommendations limit focal therapy for intermediate risk patients, while it is recommended for a selected group of high-risk patients, what was your criteria?

·      Do you always do whole gland HIFU? How did you choose to offer whole gland vs. partial/focal HIFU?

·      Did you biopsy patients who had biochemical recurrence in order to obtain “biopsy-proven recurrence”? Or was it strictly based on PSA levels?

·      Hoiw did you define incontinence?

·      Why are you putting erectile dysfunction as an “ADT related” event, when it can be secondary to the sRT of the HIFU? How did you know what was it related to?

·      The comment in line 199, “HIFU seems not significantly inferior to RP or RT combined with long-term ADT” is very biased and does not reflect the fact that most patients included in these studies represent a very selected population, can´t compare survival in such different populations, this needs to be modified. And the next sentence, same, you can´t say that with the information that you are presenting (different populations, different number of patients, different patient characteristics, etc.). Are you suggesting HIFU is better than primary RP or primary RT?

·      The whole first parr of the discussion needs to be worked on, it is presenting inaccurate interpretation of the studies. Furthermore, that is not the objective of the study, you are looking at sRT after primary whole gland HIFU in high-risk PCa patients, why are you even including a discussion about primary treatment modalities? If you want to make this comparison and make the recommendations/suggestions that you are making you need to do a completely different type of analysis

·      When you discuss the different definitions of biochemical failure afer HIFU, you also have to describe the increasing use of biopsy proven recurrence in this scenario

·      The number of AEs described in your series is low compared to other studies, you need to try to provide an explanation, describe your definitions of adverse effects, and describe the limitations of your analysis for this outcome (retrospective nature of the study, how databases were created, limited number of patients, etc.). I know you describe some of these in the limitations of the study, but in the discussion you have to mention what explanations apply to each of the outomes you describe

·      In the limitations you mention on line 273 that your outcomes and AEs can be over/underestimated, how could they be overestimated? In a retrospective analysis, you miss events because you don´t follow the guidelines and strict follow up used in RCT´s, but “overestimation” is not something you would expect in this scenario, the events that you were able to register were the ones that were significant enough so that someone decided to include in the notes, not the other way around.

·      Finally, in the conclusions, I don´t think you can say that the study “confirms” the feasibility of the treatment, it suggests feasibility, but this needs to be confirmed with larger and ideally prospective studies.

·      Time of follow up needs to be mentioned in the abstract

·      You don´t mention references about other salvage options such as salvage HIFU or sRP in the discussion

·      Minor language review need to be made

Author Response

Congratulations to the authors on your present work. It is very clear that you have a wide experience with whole gland HIFU and that you have explored the limits of this technique, which is admirable. I think the present manuscript has a lot of potential but needs to be refined, moderated, and be very careful with the description of the outcomes, variables and results in order to avoid confusions to the readers. Here are some recommendations for you to consider:

Please reference Roach´s formula for pelvic node irradiation

Response: We cited the following reference in the revised version “S. Rahman, H. Cosmatos, G. Dave, S. Williams and M. Tome. Predicting pelvic lymph node involvement in current-era prostate cancer. Int J Radiat Oncol Biol Phys 2012 Vol. 82 Issue 2 Pages 906-10”.

  • You describe your PSA follow up with measurements every 3 months, how long is that frequency sustained for?

Response: Thanks for the question. PSA was checked every 3 months for 5 years, then every 6 months for the next 5 years.

  • When you mention a median follow up of 45.9, is that after the radiation? Or are you counting since the last HIFU?

Response: Thanks for the comment. The follow-up period was calculated from the first day of the treatment (SRT or ADT).

  • How was it decided to go for sRT rather than surgery for these patients?

Response: Thanks for the comment. In our institute, many HIFU patients are not candidates for radical surgery. ADT alone is not curative. Considering the oncological control, SRT+ADT was a preferred treatment option. We have added the explanation in the 2nd paragraph of the Discussion.

  • I would describe what was the reason to choose HIFU for a high-risk patient. Most of the recommendations limit focal therapy for intermediate risk patients, while it is recommended for a selected group of high-risk patients, what was your criteria?

Response: Thanks for the comment. The criteria for HIFU in our institute were (1) localized prostate cancer with stage cT1-3N0M0, and (2) patients are unsuitable candidates for surgery or reluctant to receive radiotherapy. (Added in 2.1 Patient cohort)

  • Do you always do whole gland HIFU? How did you choose to offer whole gland vs. partial/focal HIFU?

      Response: Thanks for the question. In our practice, almost all patients received whole gland ablation. However, for those with low or intermediate-risk and unilateral biopsy-proven lesion and patients strongly asking for preservation of their erectile function, subtotal ablation with nerve sparing would be done.

  • Did you biopsy patients who had biochemical recurrence in order to obtain “biopsy-proven recurrence”? Or was it strictly based on PSA levels?

Response: In the study, treatment failure after HIFU was strictly based on the PSA level. Prostate biopsy was optionally arranged for those with PSA failure. MRI and bone scan were routinely performed for clinical re-staging due to biochemical failure. Thanks for the comment. (Rephrased in the 2nd Paragraph of 2.1.)

  • How did you define incontinence?

Response: We defined it according to the CTCAE v.4. Urinary incontinence is defined as a disorder characterized by the inability to control the flow of urine from the bladder. Thanks for the comment.

  • Why are you putting erectile dysfunction as an “ADT related” event, when it can be secondary to the sRT of the HIFU? How did you know what was it related to?

Response: Thanks for the insightful comment. We rechecked the medical record and observed the erectile dysfunction might probably be related to the late radiation toxicity at the penile bulb after SRT (though we could not definitely exclude the factor of HIFU). We corrected it in the revised version. (including Table)

  • The comment in line 199, “HIFU seems not significantly inferior to RP or RT combined with long-term ADT” is very biased and does not reflect the fact that most patients included in these studies represent a very selected population, can´t compare survival in such different populations, this needs to be modified. And the next sentence, same, you can´t say that with the information that you are presenting (different populations, different number of patients, different patient characteristics, etc.). Are you suggesting HIFU is better than primary RP or primary RT?

      Response: Thanks for the valuable comment. We have re-written the first paragraph in the Discussion.

  • The whole first parr of the discussion needs to be worked on, it is presenting inaccurate interpretation of the studies. Furthermore, that is not the objective of the study, you are looking at sRT after primary whole gland HIFU in high-risk PCa patients, why are you even including a discussion about primary treatment modalities? If you want to make this comparison and make the recommendations/suggestions that you are making you need to do a completely different type of analysis

      Response: Following the previous question, we have re-written the first paragraph in the Discussion.

  • When you discuss the different definitions of biochemical failure afer HIFU, you also have to describe the increasing use of biopsy proven recurrence in this scenario

      Response: Thanks for the valuable comment. We rephrased the first sentence in the 3rd paragraph of the Discussion as “Currently, there is no firm recommendation regarding clinically relevant PSA relapse threshold post-HIFU, although the increasing use of biopsy-proven local recurrence was reported in previous studies (8, 9, 17).”

  • The number of AEs described in your series is low compared to other studies, you need to try to provide an explanation, describe your definitions of adverse effects, and describe the limitations of your analysis for this outcome (retrospective nature of the study, how databases were created, limited number of patients, etc.). I know you describe some of these in the limitations of the study, but in the discussion you have to mention what explanations apply to each of the outomes you describe

      Response: Thanks for the valuable comment. As suggested, we revised the description of this paragraph in the Discussion. Although the evaluation criteria of AE varied between our study and the other reports, we described that the lower AE in our series might be related to the advanced RT technique, race or shorter duration of ADT use.

  • In the limitations you mention on line 273 that your outcomes and AEs can be over/underestimated, how could they be overestimated? In a retrospective analysis, you miss events because you don´t follow the guidelines and strict follow up used in RCT´s, but “overestimation” is not something you would expect in this scenario, the events that you were able to register were the ones that were significant enough so that someone decided to include in the notes, not the other way around.

Response: Thanks for the insightful comment. We have removed “overestimate’ in the revised version.

  • Finally, in the conclusions, I don´t think you can say that the study “confirms” the feasibility of the treatment, it suggests feasibility, but this needs to be confirmed with larger and ideally prospective studies.

Response: Thanks for the comment. The “confirms” has been revised to “suggests”.

  • Time of follow up needs to be mentioned in the abstract

Response: Thanks for the comment. We have added time of follow-up in the revised abstract.

  • You don´t mention references about other salvage options such as salvage HIFU or sRP in the discussion

      Response: Many HIFU patients are not candidates for radical surgery. Furthermore, in our study, patients with prostate cancer were initially treated with HIFU. Repeat HIFU may increase the risk of complications and the oncological outcome remain unclear. Lastly, earlier studies have demonstrated the feasibility of SRT ± ADT after HIFU failure. We have added other salvage options in the 2nd paragraph of the Discussion.

  • Minor language review need to be made

 Response: Thanks for the comment. Minor language editing has been made.

Round 2

Reviewer 1 Report

The manuscript was precisely corrected.